# Changes in Phenotypes and DNA Methylation of In Vitro Aging Sperm in Common Carp *Cyprinus carpio*

**DOI:** 10.3390/ijms22115925

**Published:** 2021-05-31

**Authors:** Yu Cheng, Pavlina Vechtova, Zoltan Fussy, Jan Sterba, Zuzana Linhartová, Marek Rodina, Vladimíra Tučková, David Gela, Azin Mohagheghi Samarin, Ievgen Lebeda, Miaomiao Xin, Songpei Zhang, Deepali Rahi, Otomar Linhart

**Affiliations:** 1South Bohemian Research Center of Aquaculture and Biodiversity of Hydrocenoses, Research Institute of Fish Culture and Hydrobiology, Faculty of Fisheries and Protection of Waters, University of South Bohemia in České Budějovice, Zátiší 728/II, 389 25 Vodňany, Czech Republic; ycheng@frov.jcu.cz (Y.C.); linhartova@frov.jcu.cz (Z.L.); rodina@frov.jcu.cz (M.R.); vtuckova@frov.jcu.cz (V.T.); gela@frov.jcu.cz (D.G.); mohagheghi@frov.jcu.cz (A.M.S.); ilebeda@frov.jcu.cz (I.L.); xinmiao1206@126.com (M.X.); szhang@frov.jcu.cz (S.Z.); drahi@frov.jcu.cz (D.R.); 2Faculty of Science, Institute of Chemistry and Biochemistry, University of South Bohemia in Ceske Budejovice, Branisovska 1760, 37005 Ceske Budejovice, Czech Republic; vechtova@prf.jcu.cz (P.V.); zoltan.fussy@gmail.com (Z.F.); sterbaj@prf.jcu.cz (J.S.); 3Biology Centre of Academy of Sciences of the Czech Republic, Institute of Parasitology, Branišovská 31, 37005 České Budějovice, Czech Republic; 4College of Life Science, Northwest University, Xi’an 710069, China

**Keywords:** fish, common carp, sperm aging, epigenetics, DNA methylation, sperm quality, milt, fertilization, sperm storage

## Abstract

The purpose of the current study was to analyze phenotypic and functional characteristics of common carp (*Cyprinus carpio*) spermatozoa during in vitro aging and to investigate whether global DNA methylation is affected by sperm aging. Milt was collected from five individual males, stored in vitro on ice in a refrigerator for up to 96 h post stripping (HPS) and used to fertilize eggs with intervals of 1, 24 and 96 h. Computer-assisted sperm analysis and a S3e Cell Sorter was employed to determine the spermatozoa phenotypic characteristics (motility, velocity, concentration and viability). In addition, pH and osmolality of the seminal fluid and the capacity of the spermatozoa to fertilize, hatching rate and health of the resulting embryos were examined at different aging times. Whole-genome bisulfite sequencing was used to compare the global and gene-specific DNA methylation in fresh and aged spermatozoa. The results demonstrated that spermatozoa aging in common carp significantly affects their performance and thus the success of artificial fertilization. The methylation level at the cytosine-phosphate-guanine (CpG) sites increased significantly with 24 HPS spermatozoa compared to the fresh group at 1 HPS and then decreased significantly at 96 HPS. A more detailed investigation of gene specific differences in the DNA methylation was hindered by incomplete annotation of the *C. carpio* genome in the public databases.

## 1. Introduction

Fish spermatozoa quality depends on the breeding season, age of the males and manipulation of the gametes in vitro [1,2]. Declining spermatozoa quality is directly associated with a decrease in spermatozoa performance resulting in poor fertilizing ability and the quality of the resulting offspring [3,4]. Moreover, a decline in sperm quality leads to a decrease in energy stores and changes in proteins [5]. Alterations in DNA and membrane integrity, and mitochondrial membrane potential have also been noted [2,6,7,8,9]. Sperm aging is accompanied by decreased spermiation, sperm motility and velocity, as well as modification in seminal plasma composition, and oxidant and antioxidant level changes [6,10]. The mechanism of sperm aging is thus important in reproduction. The free radical theory of aging [11] suggests that oxidative stress due to excessive free radicals or reactive oxygen species is responsible for the aging process [12]. However, recent studies indicate that oxidative stress is probably not the main promotor of the aging process in fish oocytes [13]. Epigenetics including DNA methylation are suggested as one of the important mechanisms behind the aging process [14].

DNA methylation and demethylation patterns are established by DNA methyltransferase (DNMTs) enzymes, that contain DNMT1 and DNMT3 with two distinct functions [15]. DNA methylation in spermatozoa has been related to sperm quality. For example, the level of sperm DNA methylation has a significant positive correlation with male reproductive capacity in striped bass (*Morone saxatilis*) [16] and sperm concentration and motility in humans [17]. There are many studies on DNA methylation profiles in mammalian sperm [18,19,20,21,22]. Abnormalities of DNA methylation caused by modifications in cytosine phosphate guanine (CpG) islands and shore regions result in increased susceptibility to disease, abnormalities and disorders in the offspring from more aged mice sperm [23]. Mice sperm aging causes high risks of aberrant glucose metabolism and the development of autism spectrum disorders and behaviors in intergenerational and transgenerational offspring [18]. Sperm DNA methylation in mice has also shown to be responsive to abnormal/functioned physiology [19,20]. However, some other studies reported that no imprinted genes of DNA methylation were detected in sperm of obese human [21] and cryopreserved sperm in human [22].

A fish is a suitable model animal for conducting a study on sperm aging because it displays a vast diversity of reproduction modes, and the milt contains about 1000 times more spermatozoa per ml than mammals [24]. Obtaining a large number of spermatozoa can be synchronized by modulation of the external environment such as hormonal treatment. The additional advantage is that fish spermatozoa are immotile at the time of collection, allowing easier investigation of aging over a long-time period of up to several days of storage. The motility of spermatozoa is activated only after exposure to water or suitable activating media [25]. We chose common carp for this study because it produces a large number of gametes within a single reproduction season, i.e., around 2 kg of eggs (1.6 million eggs) and 30 mL of sperm (600 × 10^9^ spermatozoa) can be produced by an individual female or male, respectively [26,27].

The aim of the present study was to monitor the changes in sperm phenotypic parameters (motility percentage, velocity, concentration, viability; pH and osmolality of seminal fluid; fertilization, hatching and malformation rates) and its relation to DNA methylation levels in common carp during in vitro spermatozoa aging. We hypothesized that sperm aging induces changes in sperm DNA methylation levels and other relevant sperm kinetic functions and fertilization parameters. Our present study is the first report that allows a better understanding of possible epigenetic alterations during the aging of common carp spermatozoa.

## 2. Results

### 2.1. Effects of Pooled Sperm In Vitro Aging on Different Phenotypic Characteristics

Spermatozoa motility measured at 15 s post sperm activation (PSA) decreased rapidly from 88.1% at 1 h post stripping (HPS) to 61.0% at 24 HPS, then to 59.5% at 48 HPS and finally 36.9% at 96 HPS (Figure 1A). Once the percentage motility of spermatozoa was analyzed in detail, focusing on fast, medium, slow and static spermatozoa (vibration in-situ) (Figure 1B), 81.7% of fast spermatozoa were recorded at 1 HPS. Subsequently, there was a sharp decrease to 49.8% of fast spermatozoa after 24 HPS and 23.9% after 96 HPS.

During milt storage, a relatively significant spermatozoa loss was found (Figure 1C). At 1 HPS, spermatozoa concentration was 27.0 × 10^9^/mL spermatozoa. Subsequently, there was a reduction to 20.3 × 10^9^/mL spermatozoa after 24 HPS and finally after 96 HPS only 14.8 × 10^9^/mL of spermatozoa remained in the milt. Figure 1C is supplemented by values of the number of motile spermatozoa from the concentration of spermatozoa per ml, which showed a rapid decrease in the number of motile spermatozoa per ml. Initially, 23.8 × 10^9^ motile spermatozoa per ml were recorded. Thereafter, there was a sharp decrease to about half after 24 HPS, and then only 11.5% of the motile number of spermatozoa per ml remained (5.5 × 10^9^ motile spermatozoa per ml) after 96 HPS.

The curvilinear velocity of spermatozoa also gradually slowed with increasing storage time (Figure 1D). The reduction in sperm curvilinear velocity (VCL) was not as abrupt as that of the sperm motility rate. At 15 s PSA, spermatozoa VCL was 148.8 µm/s at 1 HPS, then VCL decreased to 131.4 µm/s at 24 HPS. There was a significant decrease in VCL to 95.6 µm/s after 48 HPS and curiously an increase in VCL of spermatozoa to 112.0 µm/s after 96 HPS was noted. Relatively high levels of slow and moderately slow spermatozoa (medium speed) occurred at 48 HPS compared to other HPS (Figure 1B). In the case of straight-line velocity (VSL), a similar trend of a decrease up to 48 HPS with a following increase at 96 HPS was noted.

The highest value of sperm viability with 92.0% live spermatozoa was observed in fresh milt (1 HPS), and a significant decrease in live spermatozoa occurred at 24 HPS (68.3%). Subsequently, the sperm viability stayed stable around 75.83 ± 1.07% live spermatozoa at 48 and 96 HPS (Figure 1E).

The pH and osmolality of the seminal plasma also varied significantly through the milt storage time (Figure 1F,G). The change in pH was significant, decreasing from the initial value of 8.2 at 1 HPS to 7.6 and 7.8 at 24 and 96 HPSs, respectively. In contrast, the osmolality increased significantly from 268.0 mOsmol/kg at 1 HPS to 275.3 mOsmol/kg at 96 HPS.

### 2.2. Effects of Sperm In Vitro Aging from Individual Males on Phenotypic Characteristics

Sperm from the five individual males was used to confirm the validity of the phenotypic parameters achieved in the previous experiment in the pooled aging milt (Figure 2A–F). Motility of spermatozoa rapidly decreased from the initial value of 89.3% at 1 HPS to 28.2% at 96 HPS (Figure 2A). The proportion of fast-moving spermatozoa was measured as *c* 80.5% in the freshly collected milt, while at 96 HPS, only 17.7% was fast-motile spermatozoa (Figure 2B). In addition, sperm concentration showed a sharp decrease from the initial value of 23.8 × 10^9^/mL spermatozoa to only 16.3 × 10^9^/mL after 96 HPS, which represented a decrease in sperm concentration per ml of 31.2% (Figure 2C). The number of motile spermatozoa per ml decreased even more quickly, from 21.1 × 10^9^/mL to <4.8 × 10^9^/mL at 96 HPS (Figure 2C). The VCL and VSL of spermatozoa similarly decreased from values of 134.6 µm/s and 110.4 µm/s at 1 HPS to 94.2 µm/s and 74.5 µm/s at 96 HPS, respectively (Figure 2D). The proportion of spermatozoa activity with the medium and slow speed groups (Figure 2B) was similar at different HPSs, therefore the sperm velocity showed a balanced decrease (Figure 2D).

At the beginning of the experiment, seminal plasma had a pH of 8.52 and at the end, after 96 HPS, it still kept a relatively high pH value of 8.21 (Figure 2E). Additionally, osmolality increased significantly with time, increasing from 267.2 mOsmol/kg at 1 HPS to 274.5 mOsmol/kg after 96 HPS (Figure 2F).

### 2.3. Effects of Pooled Sperm In Vitro Aging on Fertilization, Hatching and Larval Malformation Rates

The results from the freshly collected sperm at 1 HPS showed that 31,250,000 spermatozoa were sufficient to obtain 79.9–97.5% fertilization and 57.4–91.9% hatching rates (Figure 3A,B). The fertilization and hatching rates of the aging sperm at 24 HPS decreased by 30.6% compared to the control at same HPS. The number of 31,250,000 sperm stored for 24 h was able to ensure fertilization and hatching at the level of 66.9% and 51.8%, respectively. Sperm fertility decreased rapidly at 96 HPS. Compared to the controls at the same HPS, fertility and hatching rates were reduced 75.7% and 71.9% at 96 HPS, respectively. Malformation rates of embryos from 24 and 96 HPS aged sperm groups (0–9.12%) were lower or not significant compared with control groups at the same HPS (Figure 3C).

### 2.4. Bisulfite Sequencing of Carp Spermatozoa Genomic DNA during Aging

The DNA methylation patterns in freshly collected and aged carp sperm were investigated at 1, 24 and 96 HPS. We performed the whole-genome bisulfite sequencing (WGBS) of genomic DNA of the three samples, each represented by three biological replicates, and we received on average 8.67× sequencing coverage of the carp genome assembly GCA_000951615.2 (1.7 Gb). The WGBS sequencing statistics and quality filtering statistics are listed in the Appendix A.

Trimmed reads were mapped to the reference using the Bismark Bisulfite Read Mapper with an average 71% mapping efficiency. Bismark PCR deduplication script detected and removed on average 18% of the mapped reads. The details of the mapping and deduplication statistics are given in Appendix A. Context-dependent methylation calls were extracted using a Bismark methylation extractor. Cytosine methylation calls were extracted for different nucleotide contexts and methylomes of all three libraries were compared. The details of cytosine methylation calls are given in Appendix A. The ratio and extent of cytosine methylation in different nucleotide contexts show values typical for a vertebrate genome with the prevalence of CpGs methylation (Figure 4) and thus the differential methylation was done with methyl-CpG (mCpG) context only. Interestingly, the cytosine methylation report revealed that the level of mCpG fluctuates at 24 HPS aged spermatozoa (Figure 4) in comparison to 1 and 96 HPS.

#### 2.4.1. Differential Methylation Analysis

Methylation calls of the three libraries were submitted for a differential methylation analysis by Defiant using pairwise comparison of 1 HPS and 24 HPS libraries, and 1 HPS and 96 HPS libraries. The details of the Defiant run are listed in Appendix A. A total of 1305 and 1729 differentially methylated regions (DMRs) were detected by Defiant in 24 and 96 HPS libraries, respectively. These two comparisons aimed at describing the gradual changes in cytosine methylation over the 4 days of carp sperm storage, using the 1 HPS library as a reference for the calculation of DMRs in both 24 and 96 HPS libraries. The number of mCpGs increased by 1.76% (*p* = 0.00376) at 24 HPS DNA from 78.9% at 1 HPS to 80.7% at 24 HPS and decreased to 78.7% at 96 HPS, which represents only an insignificant decrease of only 0.27% (*p* = 0.32257) in comparison to the 1 HPS library. This interesting DMR fluctuation over the time of sperm aging encouraged us to calculate the pairwise comparison of mCPG counts for 24 and 96 HPS libraries using the 24 HPS library as a reference and the 96 HPS library for DMR inference. In this comparison we observed a significant decrease of mCpGs by 1.97% (*p* = 0.00001) (Figure 4).

Closer investigation of the DMRs fluctuations revealed that 38 DMRs that underwent methylation at 24 HPS were found unmethylated at 96 HPS while only 3 DMRs experienced the opposite process, i.e., they were unmethylated at 24 HPS and methylated at 96 HPS. The identity of these DMRs is given in Appendix A.

#### 2.4.2. Functional Annotation of Differentially Methylated Loci

Prior to functional annotation, a re-annotation of DMRs in both 24 and 96 HPS libraries was done in order to populate an incomplete annotation record of the carp genome and identify missing transcripts and protein IDs for chromosomal loci that are actively transcribed. Additionally, missing protein IDs for transcript IDs already present in the carp genome were also collected. A summary of DMR statistics is given in Table 1.

A complete list of DMRs, including statistically supported counts of DMRs calculated by Defiant and their annotation records are given in Appendix A for pairwise comparisons of 1 × 24, 1 × 96 and 24 × 96 HPS libraries, respectively.

Functional annotation and GO enrichment of DMRs at 24 and 96 HPS were performed in order to unveil a potential linkage of CpG methylation and sperm aging. A direct link of CpG methylation with groups of genes employed in specific cellular processes suggests a regulatory function of CpG methylation in the deterioration of spermatocyte viability and/or motility. Unique protein IDs for DMRs from 24 and 96 HPS libraries were submitted to GO annotation with InterProScan and the assigned GO terms were further used for functional enrichment by GOATOOLS. Annotations could be assigned to 285 and 449 unique protein IDs from the 24 and 96 DMR subgroups, of which 186 and 281 were assigned a GO term, respectively. Only a single GO term was found significantly enriched after a false discovery rate correction in the 24 HPS dataset.

## 3. Discussion

The results obtained demonstrated that the quality of spermatozoa in common carp is significantly affected by the time of in vitro storage. The motility rate and the live spermatozoa notably decreased within the first 24 HPS together with pH of the seminal fluid. Fertilization rate also decreased from 90% to 67% at 24 HPS. These results are consistent with previous studies on other fish species, although there is considerable variation in storage capacity of sperm between species. For example, the rainbow trout (*Oncorhynchus mykiss*) sperm could be stored for 34 days [28], while zebrafish (*Danio rerio*) sperm stored in Hank’s balanced salt solution were viable for <1 day [29].

Sperm motility is considered to be one of the most useful parameters to determine sperm quality, which is highly correlated with fertilization success [30]. In the present study, a sharp decrease in sperm motility was observed at 24 HPS for pooled sperm. Such a sharp decreasing trend in sperm motility was also recorded in the experiment using individual male spermatozoa. The lowest motility rates were recorded as 36.75% and 28.23% at 96 HPS for both the pooled and individual sperm storage experiments, respectively. In addition to the reduction in sperm motility and velocity, a decrease in total spermatozoa concentration was observed with increasing storage time. After 96 HPS, only 55% of the original total spermatozoa number in ml remained in the pooled sperm and only 69% in the sperm from individual males. However, these data do not usually appear in the fish spermatology literature. Only Dietrich et al. [31] reported a similar result in common carp sperm with a relatively large reduction of spermatozoa number per ml after 72 HPS. The common carp spermatozoa die and disintegrate with increasing milt storage time. In our experiment, the highest changes in seminal plasma after 24 h were recorded with a significant decrease of pH and also with increasing osmolality.

Not only the number of spermatozoa but also the number of motile spermatozoa were significantly affected by the storage time. At 96 HPS, only 20% of spermatozoa remained motile. It was also found that fast spermatozoa predominate at 1 HPS and the most spermatozoa with static movement were detected at 96 HPS. Based on these results, at 1, 24, 48 and 96 HPS about 16–19 × 10^9^, 6 × 10^9^, 4–7 × 10^9^ and 1–1.4 × 10^9^ spermatozoa within the fast movement category were recognized, respectively, for both mixed and individual sperm experiments (if a reduction in sperm count is taken into account). The velocity and motility of spermatozoa movement determines the success of sperm competition and fertilization [24,25,32]. In a previous study [33], high correlations were found between these parameters and fertilization and hatching rates, Similar results have also been reported in Atlantic halibut (*Hippoglossus hippoglossus* [34]) and common carp [35]. Usually in freshwater bony fishes the spermatozoa motility and the time of closure of the micropyle in the egg is very short. Therefore, very fast-moving spermatozoa are pre-requisite, so that they can fertilize the egg within 20–30 s [32]. In common carp, the spermatozoa can move for a maximum of 2 min. Nevertheless, rapid movement has been recorded only up to 30 s [26,36]. A previous study conducted on Indian major carp orangefin labeo (*Labeo calbasu*) have also observed a similar significant decrease in sperm motility during the first 12 h of storage [37]. The energy source [adenosine triphosphate (ATP) is the most common] produced and stored during the quiescent stage of spermatozoa in seminal plasma has been implicated as the primary source of immediate energy needed for spermatozoa motility [38,39]. Reduction in sperm motility is assumed to be a result of a decrease in ATP and oxygen during the storage time [40]. Motility decrement depends on the impairments in various other aspects of the cell, such as the physiological state of the mitochondria, plasma membrane and DNA integrity and flagellum structure [24,32].

The fertilization rate was recorded above 90% in the fresh control groups at 24 and 96 HPS and around 20% in the aging group at 96 HPS when over 450,000 spermatozoa per ova were used. The rapid decrease observed in fertility after 96 HPS could be due to a sharp decrease in the number of fast-moving spermatozoa. Out of the total of 450,000 spermatozoa, 250–350,000 spermatozoa with fast movement were found in fresh sperm. However, at 96 HPS, the number of fast-moving spermatozoa was reduced to 20,000 from the rest of the total (see reduction in Figure 1A–C).

The motility of cyprinid sperm indicates their fertilization capacity [41]. In our present study, the velocities (VCL, VSL) of stored sperm also gradually slowed down with increasing storage time but not as fast as the sperm motility rate. Therefore, we verified the theory given by Taborsky [42] that sperm traits such as swimming speed and motile sperm are one of the factors which results in reduction of fertilization during sperm aging [42]. In practice, it is possible to compensate for the reduction of fertility due to a decrease in the percentage with good sperm velocity and motility by increasing sperm quantity. If good fresh sperm (motility above 80%) is stored over 96 h, around 20 times more spermatozoa would need to be used from the initial level. Therefore, to improve the fertilization rates after 96 HPS, a sperm use level of at least 10^6^ spermatozoa will be required.

Common carp under natural conditions fully spermiate during June to September, when the water temperature of the ponds reaches 20–21 °C. Under natural conditions, the volumes of milt produced by the males are increased by the presence of ovulating females in the ponds [43,44]. The best quality was observed in samples obtained at the middle of the natural spawning season [44,45,46]. However, some spermiation can be obtained with fish kept isolated in experimental tanks if the water temperature is above 18–20 °C. In that situation, hormonal stimulation significantly increases the volume of milt available (five to ten-fold) and ensures sperm production [47,48]. Studies on the motility parameters reflecting carp sperm qualities (concentrations of spermatozoa and percentages of motile cells) indicated that the quality of sperm obtained without hormonal stimulation is superior to that of samples obtained after hormonal injection. The sperm we subjected to total DNA methylation showed us that the level of mCpG fluctuates in 24 HPS aged spermatozoa (Figure 4) in comparison to 1 and 96 HPS. Rather, the largest changes were expected after 96 HPS. In addition, the increase in DNA methylation negatively correlates with parameters used for the evaluation of spermatozoa quality. The observation is consistent with numerous studies that put into context aberrant sperm quality parameters and changes in mCpG levels in aged and/or infertile males [17,49,50]. Surprisingly, the majority of these studies, which also include other vertebrate species, observed an increase [51,52,53,54,55] rather than a decrease [23] in global DNA methylation. The DNA methylation change after 24 HPS can be explained in relation to the environmental changes that have occurred by the spermatozoa. As mentioned above, sperm volume increased enormously after hormonal injection with varying levels of spermatozoa quality being released from the cysts in the testes. In addition to young spermatozoa, there were also older and old populations of spermatozoa. The existence of different populations with possible separation in sperm has been demonstrated in carp [56]. Stripping of sperm into the external environment and storage in vitro at 0–2 °C resulted in a new physiological state that caused a change in DNA methylation in a portion of the population at approximately 24 HPS. However, sperm with changes in DNA methylation soon died and were no longer detectable after 96 HPS. This was evident by a sharp reduction in the sperm count after 96 HPS, including pH stabilization.

Research related to fish breeding management has also greatly advanced and the search for a good molecular diagnostic marker describing fish sperm quality is now being recognized and more appreciated [57,58,59]. The data collected in our study were used to compare epigenetic changes in carp sperm DNA with established sperm phenotypic parameters in fresh and aged carp sperm. This comparison gave us an opportunity to identify genomic loci whose hypermethylation is typical for aberrant sperm populations and hence have a potential to be used as epigenetic markers. This also facilitates the diagnostics of low-quality carp sperm and introduces improved techniques for sperm storage and handling in carp breeding programs. Similar research was conducted in a previous study focused on the correlation of DMRs in high-fertility and low-fertility striped bass individuals [16]. The comparison of the two study groups revealed 171 DMRs and functional annotation of these DMRs uncovered two important groups of genes that are related to the striped bass fertility performance. In our study, we performed the GO enrichment of annotated DMRs, however, no enrichment was found in either of the study groups that were compared. It is important to note that the design of our present study and the study conducted on the striped bass are not really comparable. Additionally, the global methylation analysis in the study of striped bass was done using the Methyl-CpG binding domain sequencing (MBD-seq) method while our study used the WGBS method. It is interesting to note, however, that both low fertile spermatozoa in the striped bass and low-quality aged spermatozoa in carp are both associated with changes in DNA methylation levels. The study of the DMRs in low fertile striped bass found a clear link between the DMRs identified and the genes involved in fish fertility. In our study, we did not find any functional enrichment. One of the major limitations of our findings is the lack of a well annotated genome, which could cause the omission of important genes that would otherwise contribute to the possible enrichment in certain functional group of genes. It is thus of major importance to perform a more thorough annotation of the DMRs in the future.

On the contrary, the study of the methylome in bad and good breeders in zebrafish showed no difference in methylation levels between the two study groups [59]. However, it is important to note that the analysis of differential methylome in this study was conducted using the methylation specific restriction digestion method which is by orders of magnitude less sensitive and specific then WGBS or MBD-seq and does not provide any information about methylation status of specific CpGs. The locus specific analysis of methylation was done in the promoter of the *dmrt1* gene, which is a key regulator of male sex determination in vertebrates and is particularly useful in fishes that lack sex determination by distinct sex-specific chromosomes [60]. Similarly, this locus specific methylation analysis did not show any significant difference in methylation between good and bad breeders of zebrafish. Due to the inconsistency in study designs as well as in methodological approaches, these three studies cannot be fully compared, and more extensive and thorough research need to be conducted in order to make any valid conclusions about the methylation as a reliable marker of fish sperm quality and the effect of sperm aging.

The effect of altered DNA methylation profiles in aberrant spermatozoa are often linked to the abnormal differential expression that occurs during spermatogenesis. Since the mature spermatozoa are transcriptionally inactive due to the highly condensed state of their chromatin [61], it is possible that the hypermethylation during sperm aging is driven by the set of transcripts carried by a mature spermatozoon, whose abundance was changed during spermatogenesis. This aspect has also been addressed by studies investigating the transcript identity and abundance during spermatogenesis in mature spermatozoa [62,63].

It is hypothesized that the representation and abundance of transcripts, including transcripts for methyltransferases, could be influenced by genetic failure or external stress stimuli during spermatogenesis [64]. Poor quality parameters and lowered viability of spermatozoa are often associated with environmental stress that is believed to be manifested in the observed phenotypes via epigenetic regulation. The effect of environmental changes on the altered gene expression are largely implemented by the changes in the epigenome, which further interacts with a networks of DNA methylation reader proteins playing a role in a cascade of gene expression regulation. This assumption led to the identification of a sperm-specific catalogue of transcripts and characterization of gene expression profiles in normal [65,66] and infertile sperm in vertebrates which yielded several promising candidate transcripts that are currently in use as markers of male sperm quality and fertility [50,62,67,68,69]. The methylation changes that occurred in the subvital and aberrant sperm cells at 24 HPS might have been driven by the changes in the gene expression and thus by altered representation of transcripts that could have led to the observed hypermethylation with subsequent deterioration of sperm quality traits and eventual death of these cells.

Another factor that could play a role in the observed DNA hypermethylation with subsequent death of aged sperm cells could be the sperm sample handling procedure and subsequent storage conditions. Upon stripping, the milt samples were transferred directly into ice, which represents a 20 °C difference in comparison to the body of the carp males. Such a change in the environmental temperature could be considered a shock for the sperm cells and could thus contribute to the induction of methylation changes in the part of the susceptible sperm cell population. The mechanism driving these changes in response to environmental stress remains to be elucidated. The effect of sperm cryopreservation on the differential methylation changes has been investigated before in zebrafish [70]. Interestingly, specific regions of zebrafish genome were hypermethylated, but others were not affected. Additionally, the follow up study found that only subpopulations of spermatozoa experienced hypermethylation while others were hypomethylated.

Our data, however, require further investigation. A more thorough carp genome annotation should allow identification of other types of genomic features such as gene promoters and genomic repeats which are both frequent targets of cytosine methylation and often play a crucial role in gene expression, genome architecture or silencing of transposon activity.

## 4. Material and Methods

### 4.1. Ethics Statement and Animals

The study was conducted at the Faculty of Fisheries and Protection of Waters (FFPW), University of South Bohemia in České Budějovice, Vodňany, Czech Republic. The facility has the competence to perform experiments on animals (Act no. 246/1992 Coll., ref. number 16OZ19179/2016–17214). The expert committee of the Institutional Animal Care and Use Committee of the FFPW approved the methodological protocol of the current study according to the law on protecting animals against cruelty (reference number: MSMT-6406/2019-2).

### 4.2. Experimental Fish

The research was carried out at the Research Institute of Fish Culture and Hydrobiology of the Faculty of Fisheries and Protection of Waters, University of South Bohemia, Czech Republic. Broodstock handling was performed according to Linhart et al. [71]. Before injection and gamete collection, the males and females were anaesthetized in a solution of 2-phenoxyethanol (1:1000). The best quality eggs (darker coloration, without excessive ovarian fluid and no apparent decomposition) were visually selected from individual females and immediately used for the fertilization assay.

### 4.3. Experimental Design

After collection, spermatozoa were cooled down at 0–2 °C and motility was evaluated microscopically; sperm samples with a minimal motility of >90% were chosen for all experiments. Experimental Eppendorf tubes (Sigma-Aldrich, Taufkirchen, Germany) used for storage of milt were kept as 1200 µL of air and 600 µL of milt (ratio air: milt = 2:1) and the lids kept closed.

#### 4.3.1. Phenotypic Parameters of Pooled Sperm during In Vitro Aging

Milt from five males from a recirculation aquaculture system (RAS) was pooled, distributed in four 2 mL tubes and stored on ice in a refrigerator under aerobic conditions for 1, 24, 48 and 96 h. The motility, velocity, viability and concentration of stored spermatozoa, and pH and osmolality of seminal fluid were recorded in triplicate at each studied time point (1, 24, 48 and 96 HPS).

#### 4.3.2. Phenotypic Parameters of Sperm from Individual Males during In Vitro Aging

Milt from five individual males from RAS was distributed in fifteen 2 mL tubes (three per male) and stored on ice in a refrigerator under aerobic conditions for 1, 48 and 96 h. The motility, velocity and concentration of stored spermatozoa as well as pH and osmolality from seminal fluid were recorded three times at 1, 48 and 96 HPS.

#### 4.3.3. Fertilization and Hatching Level with Malformation Embryos Rates from Pooled In Vitro Aged Sperm

The milt samples from five males from a pond were pooled and stored on ice in three 2 mL tubes under aerobic conditions. Fertilization experiments with pooled milt were performed at 1, 24 and 96 HPS. Freshly ovulated eggs were pooled from three females for each milt storage time of 1, 24 and 96 HPS. In addition, pooled freshly collected control milt from three males from a pond were used to test fertilization at 24 and 96 HPS. Prior to fertilization, the concentration of pooled sperm from short-term storage at 1, 24 and 96 HPS was 19.06 × 10^9^ spermatozoa per ml, and pooled fresh control sperm at 24 and 96 HPS was determined at 20.53 and 27.40 × 10^9^ spermatozoa per ml, respectively.

#### 4.3.4. Whole Genome Bisulfite Sequencing from Pooled In Vitro Aged Sperm

Milt from five males from RAS was pooled, distributed in three 2 mL tubes and stored on ice in a refrigerator under aerobic condition for sampling at 1, 24 and 96 HPS.

### 4.4. Examining Phenotypic Characteristics of Spermatozoa during In Vitro Aging

#### 4.4.1. Sperm Motility, Velocity and Concentration

Distilled water supplemented with 0.25% Pluronic F-127 (catalog number P2443, Sigma-Aldrich; used to prevent sperm from sticking to the slide) was used as an activating medium (pH 7.0–7.5), that was kept on ice during the whole experiment. Spermatozoa were activated at room temperature (21 °C) by mixing the diluted sperm sample with needles into 10 µL of the activation medium on a chamber SpermTrack-10^®^ (Proiser R + D, S.L.; Paterna, Spain) at different times of storage. The activated spermatozoa were recorded microscopically and followed the techniques described previously [72]. Computer-assisted sperm analysis included the percentage of motile sperm (%), curvilinear velocity (VCL, μm/s), straight-line velocity (VSL, µm/s) and spermatozoa rate with rapid motility (>100 μm/s), medium motility (46 to 100 μm/s), slow motility (10 to 45 μm/s) and static spermatozoa (<10 μm/s). Scales in sperm analysis software were calibrated and the base on the microscope and adaptor set. Escala X and Escala Y were both set up to 1.38 μm when using 10 × lens on a negative phase-contrast condenser microscope. Quantitative analyses of all samples were conducted in triplicate.

Sperm concentration together with the total number of sperm per male and mixed sperm from five males was evaluated at different HPS. The sperm concentration (expressed as 10^9^ spz/mL) was determined by a Bürker cell hemocytometer (Marienfeld, Germany, 12 squares counted for each male) using an optical phase-contrast condenser and an ISAS digital camera (PROISER, Spain) under an Olympus microscope BX 41 (4009). For counting the cell number clearly, the Bürker cell hemocytometer containing sperm samples was placed horizontally for about 3 min for the cells to sediment.

#### 4.4.2. Osmolality and pH of Seminal Plasma 

Sperm samples were centrifuged at 17,000× *g* at 4 °C for 15 min twice to separate the cell and supernatant (Thermo Scientific, Fresco 21; Thermo Scientific, Walthan, MA, USA). From the seminal plasma collected (supernatant), osmolality from 50 µL and pH from 250 µL samples were determined. Data for osmolality were collected in triplicates by using a Freezing Point OSMOMAT 3000 (Gonotec, Germany) and expressed in mOsmol/kg. pH values in all samples were measured at different HPS with a Hach H160 handheld meter (Hach Company, Loveland, CO, USA) which was calibrated prior to each round of measurements.

#### 4.4.3. Sperm Viability Analysis

The live:dead sperm cell ratio was determined by flow-cytometric analysis using the LIVE/DEAD Sperm Viability Kit (Invitrogen/Thermo Fisher Scientific Inc.) by S3e™ Cell Sorter (Bio-Rad, Hercules, CA, USA). Prior to running samples, quality control was performed on the S3 Cell Sorter using ProLine™ Universal Calibration Beads. ProSortTM Software was used to generate density plots: FSC (area) *v*. SSC (area) and FSC (area) *v*. FL2 (area). Before measurements, live untreated sperm (negative control) and sperm subjected to repeated storage (positive control) were used to calibrate each fluorescent channel’s sensitivity, then thresholds and finally set-up regions of interest. Sperm samples were pre-diluted: 0.5 μL of sperm sample were suspended in 2 mL of 0.9% (*w*/*v*) NaCl solution. Then, 5 μL propidium iodide (PI) (Sigma-Aldrich, St Louis, MO, USA) was added to the sperm solutions, vortexed for a few seconds and incubated for at least 15 min on ice. A minimum of 10,000 spermatozoa were analyzed using a S3e Cell Sorter. The data were processed using ProSort™ software (Bio-Rad, Hercules, CA, USA). The populations with different PI channel intensities were compared for each sample and correlated with membrane damage. Populations with high PI fluorescent signals were considered as dead cells. The percentage of live:dead sperm cells was calculated based on this ratio between populations with low and high PI fluorescence intensity [73].

### 4.5. Examining Fertilization, Hatching and Larval Malformation Rates during Spermatozoa In Vitro Aging

For the fertilization experiments, pooled in vitro stored milt at different HPS were diluted in Kurokura immobilizing solution (KIS) [35], then pooled fresh eggs (0.10 g, about 70 eggs) from three females at each HPS were inseminated by pipetting 16.45 µL milt at 1, 24 and 96 HPS with 31,250,000 diluted spermatozoa (100 µL milt + 900 µL KIS).

Prior to fertilization at 24 and 96 HPS, pooled fresh milt controls from three males were diluted with the KIS, respectively. Eggs (0.10 g = approximately 0.1 mL, about 70 eggs) were inseminated by pipetting 11.41 µL for the fertilization assays at 24 and 96 HPS with again 31,250,000 diluted spermatozoa (100 µL milt + 900 µL KIS).

Milt was not added directly to the eggs, but the pipette was inserted to the bottom of the 25 mL beaker near the eggs. This procedure was repeated four times, resulting in four beakers being placed on the shaking table (at 21 °C). The shaking table was activated for 1 min (150 rpm), and 2 mL of water was pipetted into each beaker during this period. The fertilized eggs were carefully transferred from the four beakers into four glass Petri dishes (95 mL; 9 cm in diameter and 1.5 cm in depth). Each Petri dish was filled with dechlorinated water.

Each group of four Petri dishes was kept in four small plastic boxes (13.5 cm × 10 cm × 6.5 cm), filled with dechlorinated water (300 mL) which in turn were placed in one larger plastic box (28 cm × 21.5 cm × 7.5 cm [74]). These boxes were kept in an air-conditioned laboratory at 21 °C. The eggs in each Petri dish were counted as the initial total number of eggs. Dechlorinated water was gently changed in the small plastic boxes at 48 h post-fertilization and non-developing embryos were removed; water was exchanged daily after the eye-stage (48 h post-fertilization) up to hatching. The eye-stage embryos (fertilization) were counted at 48 h post-fertilization and then the hatched larvae and malformed larvae were manually counted directly after hatching over the next 5 days of incubation at 21 °C. The fertilization rate was calculated as the ratio of eye-stage embryos at 48 h post-fertilization to the initial number of eggs per Petri dish. The total hatching rate was calculated as the ratio of all hatched larvae to the initial number of eggs per Petri dish. The malformation rate was calculated as the ratio of malformed larvae (unusual body proportions, e.g., peritoneal or heart edema, irregular body axis and head malformities) to the total number of hatched larvae per Petri dish.

### 4.6. Whole Genome Bisulfite Sequencing of Sperm

#### 4.6.1. DNA Isolation from Carp Sperm Samples

Samples were processed at 1, 24 and 96 HPS. A volume of 200 μL of each sample was transferred to new sterile microtubes and samples were centrifuged at 17,000× *g* at 4 °C for 15 min twice to eliminate the supernatants (Thermo Fisher Scientific). Pellets were used to isolate the genomic DNA using RNA blue (Top-Bio, Vestec, Czech Republic) following the protocol for DNA isolation.

DNA concentration, protein and salt contamination were determined spectrophotometrically using Nanophotometer Pearl (Implen, Munich, Germany). The quality and integrity of gDNA was further tested by restriction digestion by EcoRI followed by electrophoretic separation and visual inspection of intact gDNA and the corresponding restriction digest product in 1.5% agarose gel.

One μg of total DNA of each sample was sent for WGBS to Admera Health Biopharma Services. The bisulfite conversion was done using Zymo EZ DNA Methylation Gold Kit (Zymo Research, Irvine, CA, USA), and the sequencing library was prepared using Swift Accel-NGS MethylSeq DNA library kit (Swift Biosciences, Ann Arbor, MI, USA) producing a directional sequencing library. The sequencing was done with Illumina PE 150 Cycle (Illumina, San Diego, CA, USA).

The quality of raw sequencing data was verified with fastQC-0.11.5 [75], and sequencing adapters and low-quality reads were removed with Trimmomatic-0.36 [76] using the following parameters: ILLUMINACLIP:seqprimers.txt:2:25:10 LEADING:4 TRAILING:4 SLIDINGWINDOW:4:15 MINLEN:36.

The clean reads were mapped to the common carp genome assembly (accession number GCA_000951615.2) using Bismark Bisulfite Read Mapper v0.22.3 [77] with—score_min value L, 0–0.6. Duplicated reads were removed from the Bismark output files with the Bismark_deduplicate script.

The counts of methylated and non-methylated cytosines were retrieved with the Bismark_methylation_extractor, and a pairwise differential methylation analysis was performed in Defiant v 1.1.9 [78] with minimum differential nucleotide count 4 (-d switch), minimum number of CpN 5 (-CpN switch) and -v switch with a bh argument to print a *p*-value supported by a Benjamini-Hochberg correction for each differentially methylated region (DMR). The remaining options were set to their default values or were not used if the default value was not preset. The records of DMRs were removed from the final count table if their q value was lower than 0.05.

#### 4.6.2. Functional Annotation of Differentially Methylated Regions

Chromosomal co-ordinates in the count tables of DMR were paired with the carp genome annotation file in order to collect the corresponding protein IDs if present. When the protein ID was not available, locus, transcript or gene IDs were used to search for protein ID in GenBank using an in-house BioPython v1.78 script employing the Entrez module. The remaining unannotated DMRs were further manually annotated. Chromosomal sequences of some unannotated DMRs were used to retrieve transcript ID from common carp transcriptomic data if the percent identity of the best sequence hit was above 95%.

All proteins derived from the carp genome assembly, including the ones newly collected by the above procedure, were collectively processed with InterProScan v 5.48–83.0 with the Goterms parameter [79]. The resulting report was parsed to assign GO terms to sequences in this set. A list of all accessions provided a background for functional enrichment of DMRs, whereas lists of DMRs represented the study group for the enrichment analysis.

Functional enrichment was performed by GOATOOLS, v0.8.12 [80] with Benjamini-Hochberg false discovery rate and gene ontology file version 2021-02-01.

### 4.7. Data Evaluation and Statistical Analysis

The data distribution homogeneity of dispersion was evaluated using the Levene’s test. One-way analysis of variance (ANOVA) with an LSD test for post hoc multiple comparisons was used to test for differences among the groups between sperm motility, sperm total concentration, motile sperm concentration, velocity of spermatozoa, percentage of dead and live cells, pH of seminal plasma, osmolality of seminal plasma, fertilization rate, hatching rate and malformation rate.

The DMRs were calculated by Defiant [77] which infers the DMRs using Weighted Welsch expansion hence lending a higher significance to each of the three biological replicates with greater coverage.

Line charts of sperm motility, percentage of live and dead cells, and fertilization, hatching and malformation rates were drawn with mean ± standard deviation of the mean (S.D.). All analyses were performed at a significant level of *p* < 0.05 in R 3.3.2 [81].

## 5. Conclusions

Significant decreases in the phenotypic characteristics of common carp sperm were detected after aging for 96 h from stripping. Meanwhile, seminal fluid characteristics such as pH and osmolality gradually decreased and increased, respectively. An age-related decrease in fertilization and hatching rates by the aging of common carp sperm was observed. Aging was associated with a dynamic changes of sperm DNA methylation. An increase in mCpG was found after sperm storage at 24 HPS, in comparison to the sperm at 96 HPS whose methylation level dropped to that of freshly collected sperm at 1 HPS.

Our results contribute to the research studying the molecular basis of sperm aging in common carp, which will provide information to maximize reproductive efficiencies in an important aquaculture species. However, further studies elucidating mechanism of regulation of DNA methylation during sperm aging are needed to be able to use this information in increasing the efficiency of carp breeding management.

## Figures and Tables

**Figure 1 ijms-22-05925-f001:**
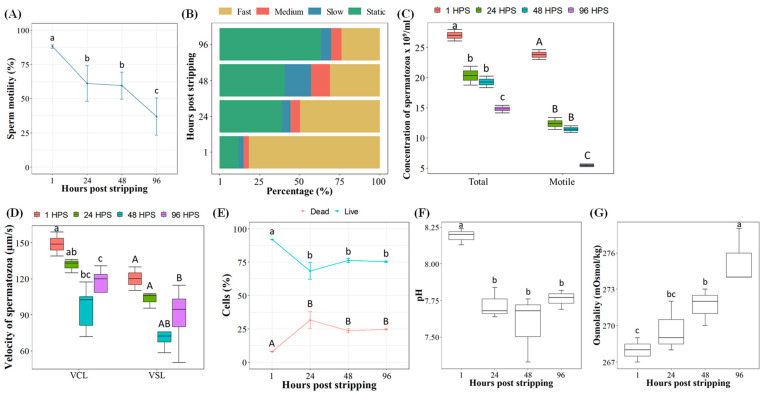
Effect of pooled milt aging (5 males) stored at 1, 24, 48 and 96 h post stripping (HPS) in common carp (mean ± S.D.): (**A**) percentage of sperm motility evaluated at 15 s of post sperm activation (PSA); (**B**) percentage of sperm motility from total motility of spermatozoa evaluated at 15 s of PSA: rapid motility (>100 μm/s), medium motility (46 to 100 μm/s), slow motility (10 to 45 μm/s) and static spermatozoa (<10 μm/s); (**C**) total and motile sperm concentration per mL; (**D**) curvilinear velocity (VCL) and straight line velocity (VSL) evaluated at 15 s PSA; (**E**) viability of spermatozoa cells; (**F**) pH of seminal fluid; (**G**) osmolality of seminal fluid. Values with different letters within lower- or upper-case letters, are significantly different [*p* < 0.05, one-way analysis of variance (ANOVA) followed by an LSD test for post hoc multiple comparisons].

**Figure 2 ijms-22-05925-f002:**
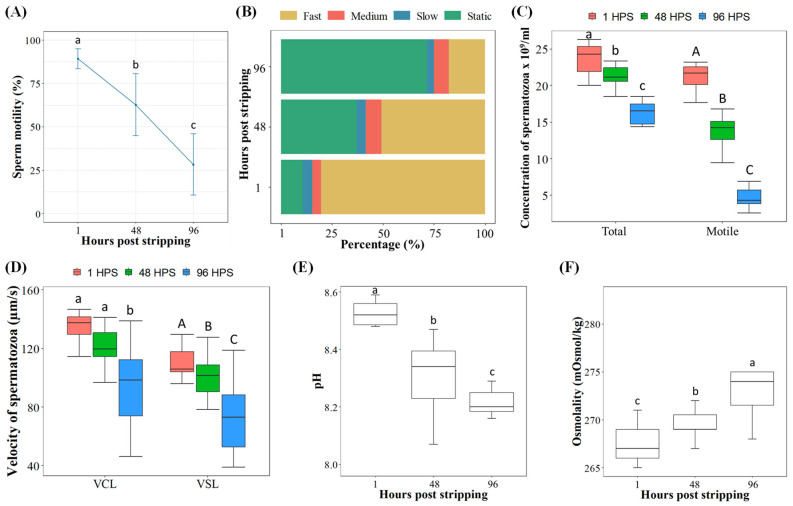
Effect of aging milt from five individual males stored at 1, 48 and 96 HPS in common carp (mean ± S.D.): (**A**) percentage of sperm motility evaluated at 15 s of post sperm activation (PSA); (**B**) percentage of sperm motility from total motility of spermatozoa evaluated at 15 s of PSA [rapid motility (>100 μm/s), medium motility (46 to 100 μm/s), slow motility (10 to 45 μm/s) and static spermatozoa (<10 μm/s)]; (**C**) total and motile sperm concentration per ml; (**D**) curvilinear velocity (VCL), straight line velocity (VSL) evaluated at 15 s of PSA; (**E**) pH of seminal fluid; (**F**) osmolality of seminal fluid. Values with different letters within lower or uppercase letters are significantly different (*p* < 0.05, one-way analysis of variance (ANOVA) followed by an LSD test for post hoc multiple comparisons).

**Figure 3 ijms-22-05925-f003:**
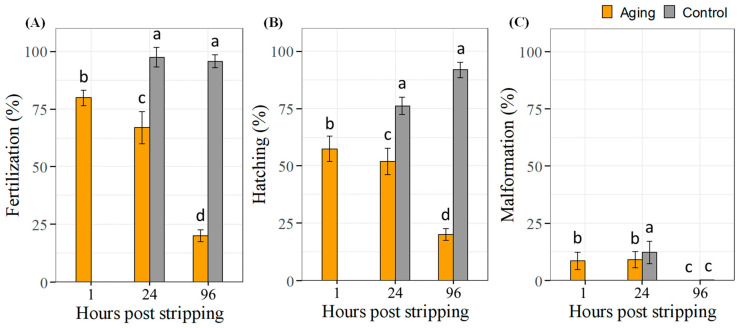
(**A**) Fertilization (eye stage), (**B**) hatching and (**C**) malformation rates (mean ± S.D.) of common carp with a total number of 31,250,000 spermatozoa. Aged milt was stored for 1, 24 and 96 h individually and pooled just prior to fertilization from four males. Controls were fresh milt collected at 24 and 96 h and used as a pool of milt from three males. Eggs were also pooled from four females. Values with the different superscript are significantly different (*p* < 0.05).

**Figure 4 ijms-22-05925-f004:**
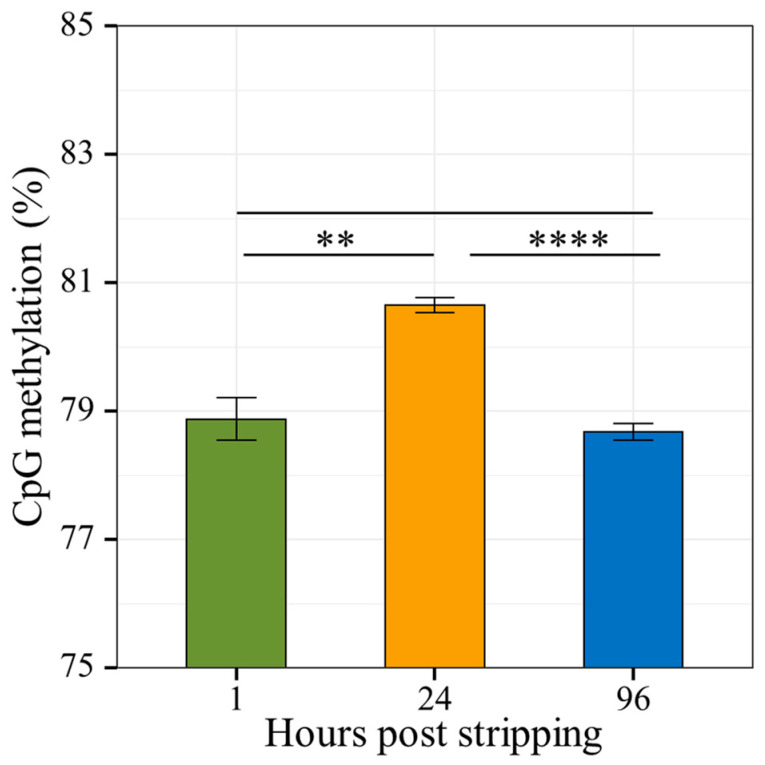
Percent change of differentially methylated regions (DMRs) among the three libraries. The DMRs were calculated by Defiant, and each value is statistically supported by three biological replicates. Asterisks denotes *p*-value between two groups (** *p* < 0.01, **** *p* < 0.0001).

**Table 1 ijms-22-05925-t001:** An annotation summary of differentially methylated regions (DMRs) in both 24 and 96 h post stripping (HPS) libraries. Only unequivocally assigned gene accessions and DMRs having a >95% sequence identity hit in the common carp transcriptomic data were considered in the list of genes. The remaining DMRs with no available evidence of expression were included in the list of putative intergenic regions.

Library	CpG Methylation State	DMRs	No. of Genes	No. of Intergenic Regions
24 HPS	methylated	920	123	797
	unmethylated	385	258	127
96 HPS	methylated	875	300	575
	unmethylated	854	288	566

## Data Availability

Not applicable.

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
