# Peer review of "Changes in Phenotypes and DNA Methylation of In Vitro Aging Sperm in Common Carp Cyprinus carpio"

_ijms, 2021, doi:10.3390/ijms22115925_

Round 1

Reviewer 1 Report

The manuscript is essentially well-written with a sound experimental design and using highly sohpisticated state-of-the-art research technology. Their use of WGBS is especially outstanding and pioneering in fish spermatology. The reviewer, however, misses at least a speculative description of the pathways affected by sperm storage after the authors took the pains (and costs) of conducting WGBS. Admission of inability to draw conclusions because carp genome is poorly annotated is not good enough. The authors do an admirable job by describing methylation changes in general, linking those to the results and their conclusions are logical, however, with such a powerful tool as WGBS, more is required.

Minor comments:
Authors should be consistent with the use of thousand separators, in most cases commas are used but in some cases spaces were found.
Rows 68-69: the sentence sounds awkward: "obese human and cryopreserved sperm". Why, cryopreserved sperm was not human? Also was the sperm obese or the human who donated it?
Row 160: There must be some sort of a mistake here. According to what is shown on the chart, there is no way that fertilization and hatching at 96 hours post stripping resulted in over 70% of its control. This is closer to 20-ish.
Row 315: The number is not written as an exponentiation but as an integer 1×106 instead of 1×106. Also, it is unnecessary to multiply it by one.

Reviewer 2 Report

The manuscript entitled: “Changes in phenotypes and DNA methylation of in vitro aging sperm in common carp Cyprinus carpio” is well written and provide new insight into the fish sperm in vitro ageing process. These findings could be of great importance for the hatchery practice.

The abstract is well written and provides basic important information regarding the content.

The introduction provides clear background for the presented research.

Materials and methods are chosen correctly and well described.

Results are clearly described.

The discussion part could be shortened and fused to sam extent. Specific comments are given in the attached manuscript.
